# Physical Education Teachers' Representations of Their Training to Promote the Inclusion of Students with Disabilities

Tadeu Celestino [1,2], Esperança Ribeiro [2], Elsa Gabriel Morgado [3,4], Levi Leonido [5,6,*] and Antonino Pereira [2]

1 Faculty of Social and Human Sciences, University of Beira Interior, 6201-001 Covilhã, Portugal; titta2323@hotmail.com
2 School of Education, Center for Studies in Education and Innovation, Polytechnic Institute of Viseu, 3504-510 Viseu, Portugal; esperancaribeiro@esev.ipv.pt (E.R.); apereira@esev.ipv.pt (A.P.)
3 Polytechnic Institute of Bragança, 5300-253 Bragança, Portugal; elsa.morgado@ipb.pt or emorgado@ucp.pt
4 Centre for Philosophical and Humanistic Studies, Portuguese Catholic University, 4710-297 Braga, Portugal
5 School of Human and Social Sciences, University of Trás-os-Montes e Alto Douro, 5000-801 Vila Real, Portugal
6 Center for Research in Arts Sciences and Technologies, Portuguese Catholic University, 4169-005 Porto, Portugal
* Correspondence: levileon@utad.pt

**Abstract:** School inclusion is based on the need to adopt and implement a holistic view of education, training, and human development embodied in the idea of everyone, for everyone. In the context of Physical Education (PE), there are still several constraints to the realization of this universal desideratum. Among these, teacher training and qualification for the inclusion of students with Specific Health Needs (SHNs) stands out. That is, students with physical and mental health problems whose impact is significantly manifested in the learning process. Thus, the objective of this study was to identify the representations of PE teachers about their training to develop inclusive processes with students with SES. Participants in this study were 151 PE teachers from different regions and districts of Portugal (Algarve, Aveiro, Castelo Branco, Lisbon, Porto, and Viseu) who had $23.6 \pm 8.1$ years of teaching service. Teachers answered an online questionnaire, on the Google Forms platform, with open and closed questions about their education and training to develop inclusive processes in PE. The results indicate two significant dimensions: (1) initial training for teaching inclusive PE and (2) continuous training for inclusion. Regarding initial training, a large majority of the teachers under study, at the end of their initial training, did not have the essential skills to teach PE to students with SES. It was also identified that a large majority reported not having had any contact with students with SES throughout their training process for teaching. It was also recognized that this training was not adjusted to the development of intervention skills with students with SHN. Regarding continuous training, it was identified that attendance at this training increased their skills to teach PE to students with SHN. Workshops/actions/training courses are the main training models adopted. However, it is recognized that the training provided does not respond concretely to their training needs to intervene with students with SHN, since teachers essentially seek to improve intervention in the context of inclusive physical education. We conclude that teacher training for inclusion is not yet fully adjusted to the reality of the inclusive school paradigm. In this sense, in practical terms, the following are suggested: (1) the need for reinforcement in study plans with specific and long-term curricular units; (2) the introduction of real practice components in context; and (3) supervised pedagogical practice in diverse contexts.

**Keywords:** teacher training; physical education (PE); inclusion in physical education; school inclusion

## 1. Introduction

The 2030 Sustainable Development Agenda of the United Nations (UN) has as one of its objectives to guarantee "quality and equitable education" for all, leaving no one behind [1]. Indeed, the adoption of a truly inclusive school is suggested, where all students

learn together, whenever possible, regardless of their differences and difficulties [2]. The diversity, complexity, and intensity of today's social situations require assertive and diverse responses from schools, particularly from teachers. For this, education professionals need to be agents of change, with values, knowledge, and attitudes that allow access and success for all students [1].

In this sense, some guidelines have emerged with an impact on teacher training programs and, very particularly, on their training to work in the context of the inclusive education paradigm [3,4]. Consequently, the need to train professionals capable of assuming responsibility for the education of all students without exception emerges [3,4]. In other words, there is a need for education professionals to develop greater awareness of the importance of inclusive philosophy for the development of societies. In fact, several authors highlight that increasing knowledge about the process of school inclusion in a higher education context represents one of the predictors of effectiveness in inclusive education [5–7].

Therefore, the key factor for the success of inclusive education lies in the need for special attention to the initial and continuous training of teachers [8,9], which enables them to respond to the characteristics and needs of diversity in schools. This is the success of inclusion rooted in the need for well-trained and prepared teachers [10,11].

The Bologna Declaration arises from the need to standardize educational policies between the various countries of the European Union. This contains the basic guidelines for the harmonization of the higher education system and, consequently, the impetus for the reform of teacher training in several countries.

In Portugal, the operationalization of this process is regulated, among others, by Decree-Law no. 74/2006 of March 24th, with the changes introduced by Decree-to-Law no. -Gram. 220/2009, and more recently by Decree-Law no. 79/2014, which changed previous teacher training models. These regulations materialized the reformulations and adaptations in teacher training curricula.

Notwithstanding, and based on recent evidence on the inclusion of students with Special Health Needs (SHNs) in PE classes, scientific unanimity is recognized in the existence of various constraints (inadequate attitudes, training deficits, perceived disability, feelings of frustration, misperceptions of disability, etc.) felt and reported by PE teachers [12–16].

Also taking these facts into account, higher education institutions, particularly those that train PE, have been readjusting their training curricula. Therefore, they are taking the opportunity to reflect on how to improve the skills of their trainees to better intervene in the current paradigm of inclusive schools [17].

However, the literature in this area of study has warned that many PE teachers consider that they are not adequately prepared to develop processes for including students with disabilities in their classes [13,15,18]. It was identified that there is a low level of intervention skills when there is a need to adapt content to its diversity [19,20]. Therefore, it was recognized that these constraints have underlying constraints at the level of specific initial and continuous teacher training [11,14,21–24].

Very recently in Portugal, the Organization for Economic Cooperation and Development (OECD) reported that 61% of teachers who teach in the 3rd cycle in public and private schools in Portugal did not feel prepared to teach students with special education needs at the end of their formative cycles. Even the area of special education appears to be one of the areas in which teachers assume they have the greatest training needs [25].

On the other hand, nowadays, inclusion in PE still seems to be hostage to the intense debate that has been dragging on between the perspectives of behaviorist training (pedagogy for performance) versus the critical perspective, denoting, still, a strong orientation towards the result and for the technicalization of PE. Therefore, paraphrasing Rodrigues, "(…) the training of teachers in special educational needs leaves much to be desired in Portugal" [26]. Effectively, this observation is in line with several studies that also show the inconsistency of the specific training developed in this context, whether in initial training courses, continuous training, or even within the scope of pedagogical practice [27].

The bibliographic research carried out allows us to observe that there is little information about the perception of Portuguese PE teachers about training for inclusion in Portugal. Consequently, questions such as the following arise: (i) What reasons are presented concomitantly to the constraints that PE teachers feel regarding their intervention with students with SHN? (ii) Are the curricula of PE teacher training courses adjusted to the current inclusive school paradigm? (iii) Does ongoing training for inclusion meet the real needs of teachers in the school context?

Thus, the objective of this exploratory study was to identify the representations of PE teachers about their training to develop inclusive processes with students with SES.

## 2. Materials and Methods

### 2.1. Participants

Taking into account the need to bring together a group of teachers with similar training and experience, we resorted to homogeneous sampling [28]. The criteria used for the selection of participants in this study were the following: (a) qualification for PE teaching and (b) teaching PE in Portuguese public schools. The final sample consisted of 151 PE teachers from various regions and districts of Portugal (Algarve, Aveiro, Castelo Branco, Lisbon, Porto, and Viseu) who had $23.6 \pm 8.1$ years of teaching service. Of these teachers, 42% ($n = 64$) taught in the 2nd cycle (which includes students aged between 10 and 13 years old), and 58% ($n = 87$) in the 3rd cycle and secondary (which includes students between 14 and 18 years of age). With regard to academic training, 59% ($n = 89$) were graduates, 37% (56) were masters, and 4% ($n = 6$) were doctors. With regard to the teaching experience with students with SHN, 96.6% ($n = 146$) have already worked with these students and the remaining 3.3% ($n = 5$) have never taught PE with this population.

### 2.2. Instruments

The elaboration of the questionnaire was based on the reference literature [29–31] in this field of study, the objectives of the study, and the methodological procedures of the development of questionnaires proposed by Glhiglione and Matalon [32]. This was submitted to a validation process, which went through several stages, namely (i) literature review and elaboration of the questions, to be submitted to four specialists (two PhDs and two masters in adapted physical education); (ii) pre-test, applied to nine PE teachers, who did not participate in the final study; (iii) resubmission to specialists for final validation; and (iv) availability on the Google Forms online platform, which generated a link that subsequently made it possible to share it via email.

In terms of structure, the questionnaire is divided into seven parts: (i) study framework, confidentiality and completion instructions, and biographical data; (ii) initial training with questions alluding to competencies to teach inclusive PE, quality and adequacy of training attended for inclusion, adequacy of initial training to the needs of teaching intervention for inclusion, perception of the current PEF training for inclusion, and suggestions for improvement (for example, "Do you consider that after completing your degree/master's degree (qualification to teach in physical education) you had the skills to teach PE to students with SHN?", "In your initial training, did you have any unit curriculum that would bring you into contact with people with SHN?", and "In your opinion, do you consider that the training you obtained in initial training was adjusted to intervene with students with SHN in PE classes?"; and (iii) continuous training with questions related to the perception of continuous training for inclusion, training carried out, suitability of continuous training to the needs of teachers, and reasons for attending inclusion training (for example, "Do you consider that after completing your ongoing training you have the skills to teach PE to students with SHN?", "Have you undertaken ongoing or specialized training in the field of special needs throughout your professional career?", and "Do you think that the ongoing training provided within the scope of the inclusion of students with special needs in physical education meets your training needs to intervene with students with special needs?").

*2.3. Procedures*

After obtaining the link to access the questionnaire, collaboration requests were sent via email to the presidents of the Associations of Physical Education Teachers (APEF) in various regions and districts of Portugal, namely Algarve, Aveiro, Castelo Branco, Lisbon, Porto, and Viseu, requesting that their associated PE teachers share it. Likewise, the same link was shared by the authors' social networks and their network of contacts, personal and professional, via email. Teachers were able to answer the questionnaire freely and informedly between March and December 2020. In this sense, teachers, before answering, expressed their informed consent in order to comply with the principles of the Declaration of Helsinki.

Given the exploratory nature of the study, we used the Google Forms platform to collect data. Using this platform allowed us to reach a greater number of teachers. However, on the other hand, it becomes a limitation given the difficulties in validating the identity of respondents and, consequently, their answers. In order to alleviate this constraint, we came into direct contact with some teachers in the authors' contact network, as well as seeking to make the questionnaire available through the APEFs' associated network.

*2.4. Data Processing*

Data treatment, taking into account the mixed nature of the study, followed two distinct but complementary approaches: (i) quantitative—statistical analysis and (ii) qualitative—content analysis of responses to open questions.

With regard to the first, descriptive statistics techniques were used, presenting the values as a percentage. Regarding the second, the content analysis technique [33] was used. The QSR NVivo11 software was used to code the interview transcripts, with the categories defined a posteriori and subject to procedures that attested to their fidelity and validity [33,34].

**3. Results**

The main objective of this exploratory study was to identify the representations that PE teachers have about their training to develop the inclusion of students with SHN in classes. Two dimensions of analysis were extracted from the data: (1) initial training for teaching in inclusive PE and (2) continuous training for inclusion.

*3.1. Training*

Initial Training Initial for Teaching in Inclusive Physical Education

As shown in Table 1, regarding the dimension of training and skills to teach students with SHN, we identified that, in terms of the perception of teachers, after leaving training courses that enable them to become teachers, the majority of this group of teachers (65.5%, $n = 99$), considered that they did not possess, after their initial training, the essential skills to teach PE to students with SHN. However, it is observed that 32.5% ($n = 49$) considered that they had these skills, and 2.0% ($n = 3$) did not express an opinion.

**Table 1.** Perception of competencies to teach students with SHN at the end of training courses.

| Does Not Reply | No Opinion | Totally in Disagreement | In Disagree-ment | Totally Disagree | Accordingly | Totally Agree | Total in Agreement |
|---|---|---|---|---|---|---|---|
| % | 2% | 25.8% | 39.7% | 65.5% | 26.5% | 5.9–6% | 32.5% |
| ($n = 0$) | ($n = 3$) | ($n = 39$) | ($n = 60$) | ($n = 99$) | ($n = 40$) | ($n = 9$) | ($n = 49$) |

With regard to the competencies in which the teachers felt more prepared, based on the analysis of Table 2, we can generally see that knowledge about the objectives of physical education, 82.8% ($n = 125$), pedagogical knowledge, 78.1% ($n = 118$), and knowledge of the teaching subject, 76.2% ($n = 115$), were configured, significantly, as being the knowledge

in which teachers felt more prepared after completing the training process of training for teaching in physical education.

**Table 2.** Perception of the competencies in which they felt more prepared after leaving teacher training courses.

| Skills | n | % |
|---|---|---|
| Knowledge of the objectives of physical education | 125 | 82.8 |
| Pedagogical knowledge | 118 | 78.1 |
| Knowledge of the teaching subject | 115 | 76.2 |
| Didactic knowledge | 94 | 62.3 |
| Methodological knowledge | 90 | 59.6 |
| Knowledge of educational aims and values | 80 | 53.0 |
| Knowledge of the curriculum | 76 | 50.3 |
| Knowledge of pedagogical content | 76 | 50.3 |
| Knowledge of the educational context | 46 | 30.5 |
| Knowledge of students and their characteristics | 41 | 27.2 |

As for the dimension of quality and adequacy of the training attended, when looking at Table 3, it appears that the vast majority (67.5%, $n$ = 102) of the teachers under study refer not to have had any contact with students with SHN throughout their teacher training process while 32.5% ($n$ = 49) claim to have had this contact.

**Table 3.** Representation of contact with students with SHN in initial training.

| Category | n | % |
|---|---|---|
| Yes | 49 | 32.5 |
| No | 102 | 67.5 |

Regarding the adjustment and adequacy of initial training to the needs of teaching intervention with students with SHN in PE classes, Table 4 ($n$ = 100) considers that the initial training he attended was not adjusted to the development of intervention skills with students with SHN. Nevertheless, 32.4% ($n$ = 49) consider that the training obtained in the PE teacher training course was adjusted to the development of knowledge to intervene with students with SHN in PE classes and 2.0% ($n$ = 3) do not express an opinion.

**Table 4.** Representation on the adequacy of the initial training obtained to intervene with students with SHN.

| Does Not Reply | No Opinion | Totally in Disagreement | In Disagree- ment | Totally Disagree | Accordingly | Totally Agree | Total in Agreement |
|---|---|---|---|---|---|---|---|
| 0% ($n$ = 0) | 2.0% ($n$ = 3) | 32.4% ($n$ = 49) | 33.8% ($n$ = 51) | 66.2% ($n$ = 100) | 20.5% ($n$ = 31) | 11.3% ($n$ = 17) | 31.8% ($n$ = 48) |

With regard to the perception of the current training of physical education teachers for inclusion, when looking at Table 5., they are not properly prepared to intervene in the context of the inclusive school. We found that 12.6% ($n$ = 19) of the teachers had a contrary perception. Nevertheless, it is important to emphasize that a majority of this group of teachers under study do not have a formed opinion (37.1%, $n$ = 56).

**Table 5.** Representation on the current adequacy of teacher training courses for the inclusion of students with SHN.

| Does Not Reply | No Opinion | Totally in Disagreement | In Disagree- ment | Totally Disagree | Accordingly | Totally Agree | Total in Agreement |
|---|---|---|---|---|---|---|---|
| 0% ($n$ = 0) | 37.1% ($n$ = 56) | 21.8% ($n$ = 33) | 28.5% ($n$ = 43) | 50. 3% ($n$ = 76) | 9.3% ($n$ = 14) | 3.3% ($n$ = 5) | 12.6% ($n$ = 19) |

With regard to suggestions for improving training courses for PE teachers, so that they improve their training response and trainees are better prepared to intervene with students with SHN, Table 6 shows us the result of the content analysis of the answers obtained.

**Table 6.** Suggestions for improving initial training courses in PE.

| Category | Number of Citations | Subcategory | Number of Citations |
|---|---|---|---|
| Units<br>Curricula | 53 | Practices<br>Specific | 14<br>39 |
| Supervised pedagogical practice in a specific context | 39 | Schools<br>Clubs | 33<br>6 |
| Contact with reality | 9 | | |
| Others | 38 | | |

Thus, the most representative categories are identified as the main suggestions, namely (i) curricular units ($n = 53$), subdivided into specific ($n = 39$) and practical curricular ($n = 14$), to identify in the following expressions of the teachers under study:

*"There will have to be greater reinforcement of the importance of subjects related to special educational needs".*

*"The existence of a specific discipline in this area in the curriculum".*

*"Have specific disciplines that address the problem".*

*"Specific subject on the various types of disabilities and their inclusion in physical education classes".*

and (ii) supervised pedagogical practice in contexts ($n = 39$), subdivided into school ($n = 33$) and club ($n = 6$), with the category of contact with people with disabilities ($n = 9$), as can be seen in the following statements by the teachers:

*"Being able to have contact in "small internships" with the reality of students with special needs, in public schools".*

*"Contact directly with special populations".*

*"I think there should be a greater focus in the courses on a practical component of intervention in the context of physical education with students with special needs, as well as training in adapted sports, where intervention methodologies are acquired depending on the specificity of the students.*

*During the professional internship, interns should be challenged to also teach classes of students with SEN. Increase the number of curricular units relating to "Special Populations" in universities and polytechnics (differentiate and expand the curriculum). Practical experiences in clubs and institutions that encourage the practice of sports for children and young people with SEN".*

*3.2. Continuous Formation*

Continuous Training for Inclusion

In relation to the dimension of continuous training and the perception that teachers have of its frequency to increase intervention skills, as we can see in Table 7, in general, a large part of the teachers under study 56.3% ($n = 85$) consider that the completion of the ongoing training carried out increased their skills to teach PE to students with SHN. The opposite perception was found in 39.7% ($n = 60$) of teachers, who admitted that they disagreed with not having deepened their skills to teach PE to students with SHN.

As shown in Table 8, with regard to the type of specialized training carried out, we found that, for 76.8% of the teachers, the participation in workshops/actions/training courses is configured as the type of training preferred, followed by the preferred training

most frequented by this group of teachers, preceded by postgraduate studies, 11.3% (*n* = 17), and master's degrees, 8.6% (*n* = 13). It should be noted that 19.2% (*n* = 29) of teachers reported not having completed specialized continuous training in the field of special needs.

**Table 7.** Perception of competencies to teach physical education with students with SHN after completing continuous training.

| Does not Reply | No Opinion | Totally in Disagreement | In Disagreement | Totally Disagree | Accordingly | Totally Agree | Total in Agreement |
|---|---|---|---|---|---|---|---|
| 0.7% (*n* = 1) | 3.3% (*n* = 5) | 14.6% (*n* = 22) | 25.1% (*n* = 38) | 39.7% (*n* = 60) | 43.7% (*n* = 66) | 12.6% (*n* = 19) | 56.3% (*n* = 85) |

**Table 8.** Preferences for the type of specialized training carried out within the scope of SHN.

| Type of Specialized Training | *n* | % |
|---|---|---|
| Workshops/actions/training courses | 116 | 76.8 |
| Postgraduate | 17 | 11.3 |
| Master's degree | 13 | 8.6 |
| PhD | 0 | 0 |
| I did not do it | 29 | 19.2 |
| Did not reply | 1 | 0.6 |

Regarding the representation that the teachers under study have of continuous training, made available within the scope of inclusion of students with SHN in PE, based on Table 9, we observe that most teachers, 67.8% (*n* = 81), consider that the training provided does not respond to their training needs to intervene with students with SHN. Conversely, however, 33.8% (*n* = 51) are of the opinion that the training provided effectively responds to their training needs to work with students with SHN within PE classes.

**Table 9.** Representation on the adequacy of continuous training provided within the scope of SHN.

| Does Not Reply | No Opinion | Totally in Disagreement | In Disagreement | Totally Disagree | Accordingly | Totally Agree | Total in Agreement |
|---|---|---|---|---|---|---|---|
| 1.3% (*n* = 2) | 11.3% (*n* = 17) | 25.8% (*n* = 39) | 27.8% (*n* = 42) | 53.6% (*n* = 81) | 30. 5% (*n* = 46) | 3.3% (*n* = 5) | 33.8% (*n* = 51) |

With regard to the motives and reasons underlying the need to carry out continuous training within the scope of the SHN, Table 10 shows us the results of the content analysis of the 119 responses made available.

**Table 10.** Reasons underlying the undertaking of continuous training within the scope of special educational needs.

| Category | Number of Citations | Subcategory | Number of Citations |
|---|---|---|---|
| Improve the intervention | 116 | Increase teaching efficiency | 45 |
| | | Specific pedagogical update | 42 |
| | | Know the problems | 17 |
| | | Know the functionality of students | 12 |
| Personal appreciation | 47 | Private interest | 18 |
| | | Professionalism | 17 |
| | | Altruism | 12 |
| Deepen knowledge | 18 | Filling training gaps | 12 |
| | | Practice | 6 |
| Others | 54 | | |

Thus, the most representative categories show that improving intervention, personal development, and deepening knowledge are the main reasons underlying the implementation of continuous training in the context of SHN, as identified in the statements of the teachers under study:

*"Increased my competence to respond more effectively; improve knowledge of the various disability conditions; Increase the number of work tools as well as my pedagogical intervention".*

*"Feeling the need to learn teaching techniques/methodologies; need to learn/know the specific characteristics of a disability that leads to a student being considered with special needs".*

*"Personal formation; academic need; enrichment of my academic tools".*

*"Improve my activity to be able to provide increasingly appropriate and effective responses".*

*"Having students with special needs; Have knowledge of students' needs".*

## 4. Discussion

This study sought to identify the representations that physical education teachers have about their training and capacity to develop inclusive processes. This analysis is pertinent given the recent regulatory framework that regulates not only teacher training but also the inclusive school paradigm.

In fact, the main results show that the dimension of training is assumed to be a central pillar of all-inclusive dynamics in the context of PE. In fact, the success of PE students, and particularly those with SHN, is, in addition to other factors, dependent on the PE teacher and very closely on their initial education and training [35,36].

However, as we found out with this group of teachers, their perception of their initial training is that, at the height of their training, this was not adjusted to the reality of inclusive education in general and the development of assertive and inclusive responses for students with SHN in a more specific way. In this sense, similar results have been reported by several studies developed in this teaching context where, effectively, the lack of specific training and the absence of specific knowledge on how to include students with SHN in PE classes are the main barriers to inclusion in PE classes [31,37,38].

The understanding of such evidence, in turn, in addition to other underlying conditions, is also echoed in the results of Celestino and Pereira [39] when characterizing the initial training of PE teachers in Portugal, whose conclusions suggest that the training that is developed appears to assume contours of a brief "sensitization" for the theme of inclusion.

Closely related to this perception, it was also verified that the most representative competencies, in which the teachers felt better prepared at the end of the training process for teaching, were the knowledge of the objectives of the discipline, the general pedagogical knowledge, and the knowledge of the teaching matter.

Taking this aspect into account, we can better understand one of the reasons underlying the more negative perspective that these teachers have regarding specific training and training to intervene with students with SHN. In fact, it appears that the didactic and methodological intervention skills that could make a difference when it comes to promoting inclusive adaptations did not appear to be the most representative at the time of the end of teacher training in this group of PE teachers.

Concomitantly, and linked to this perception, it was identified that a large majority of the teachers under study, throughout their initial training, did not have contact with students with SHN, with the damage that this aspect entails for the future consolidation of conceptions, postures, and interventional dynamics in the face of inclusion and disability being highlighted.

The literature is also unanimous in stressing that inadequate preparation for developing inclusive processes in teachers has been shown to be a negative predictor of their attitudes toward working in inclusive environments [40].

Indeed, in the Portuguese context, the Organization for Economic Cooperation and Development [25] identifies that more than half (61%) of teachers who teach in the 3rd

cycle in public and private schools in Portugal did not feel prepared to teach students with special education needs at the end of their formative cycles.

Effectively, similar results have been identified by several studies, which point out that one of the determinants for the success of inclusion in physical education emerges precisely from the need to improve the practical knowledge of teachers' skills through the development of training components of practice in context throughout the training process [41]. From this perspective, we also find evidence that teachers with more hours of contact with students with SHN in their training processes develop more positive attitudes toward inclusion [42].

In the same vein, in a study carried out by Hemmings and Woodcock [43] with student teacher candidates, they concluded that it is pertinent during teacher training processes to develop dynamics that promote real contact with the diversity of school reality.

In short, there seems to be a scientific consensus on the need to include real opportunities for contact with students with disabilities in teacher training programs [8,39,44]. In this way, there is not only the possibility of reinforcing the acquisition and consolidation of competencies for action and intervention, but also the demystification of ideas and prejudices that can only be developed through close links with practical reality and direct contact [45,46].

In short, the representations of this group of teachers suggest the need for further reflection on the paradigm of teacher training and capacity building for inclusion in PE.

With regard to the dimension of continuous training, in the context of special educational needs, it emerges as an important facilitator of inclusion in PE [41]. Likewise, as recognized by the Organization for Economic Cooperation and Development [25], the area of special education appears to be one of the areas in which teachers assume they have greater training needs and hence the relevance of their search.

These facts are also recognized in our results when we identify that the underlying reasons for the demand and attendance of this type of training are closely related to the need for teachers to improve and perfect their practical intervention. In this sense, they aim not only at specific pedagogical updating, or the need to deepen knowledge essentially of a practical nature on the various problems (disabilities), but also intend to better understand the functionality of a given problem and thus improve the response to the needs and specificities of students with SES. Thus, it is recognized that specific training of a more practical nature has been sought after, as it is an essential assistant for the teacher when developing inclusive processes within PE [41].

Nevertheless, the evidence shows a certain tendency in which teachers, firstly, preferred more informal learning moments. That is, it is recognized that there is a certain tendency to opt first for moments of learning and informal sharing, namely, the advice or support of a colleague, the establishment of sharing networks between peers, or other specialists [41] over attending structured training.

In this sense, and despite identifying a favorable opinion regarding continuous training in relation to special needs, a generalized representation is identified that the training that is made available does not respond to the more specific desires and needs of teachers to intervene with SHN students. This fact, in turn, was also recognized by Reis, Galvão, and Baptista [43], who found that the teacher training provided, in general, still followed a traditional training model and was out of context with reality. In the same sense, add the authors, this only serves for teachers to comply with the mandatory attendance that is imposed on them for their career progression.

Consequently, this aspect may justify, to a certain extent, the fact that many PE teachers autonomously seek their own knowledge, sometimes investigating and experimenting with intervention strategies, and sometimes looking for information about problems and adaptations of activities and exercises [31,47–49]. Indeed, more research is needed on this issue in order to effectively understand the impact of informal training on the development of more assertive intervention skills with students with SHN.

In short, in general, the results found here are in line with what has been reported in the specialized literature, which has been showing precisely the inconsistency of the specific training developed in this context, in particular, whether in the initial training course, continuous training, or even within the scope of pedagogical practice [27].

Likewise, it is noted that the training of PE teachers for inclusion has focused more on the "what" of the disability, rather than the "how" to include someone with a disability in physical and sports activities [50,51]. In this sense, we suggest the need to develop training plans that effectively meet teachers' real needs, such as consolidating the pedagogical and didactic dimensions of PE, as well as assertive strategies for inclusion in activities [52].

## 5. Conclusions

Taking into account the findings, we easily infer the existence of an imbalance between what is declarative knowledge and procedural knowledge in the context of specific training within the scope of special needs, which ultimately translates into a lack of pedagogical, didactic, or even methodological skills to intervene with students with disabilities and assertively implement inclusive processes in physical education classes and school sports.

According to the results, we easily corroborate and better understand the unanimous position of the teachers with regard to the need to strengthen the level of initial teacher training. Very particularly, in the reinforcement of the study plans with the need to value curricula with specific curricular units, as well as the introduction of practical components, such as the promotion of moments of supervised pedagogical practice in diversified contexts, aiming not only to deepen and consolidate the specific pedagogical, didactic, and methodological intervention skills, but also to highlight the need to promote direct contact with people with disabilities or students with SHN.

Given the above, we conclude that teacher training for inclusion is not yet fully adjusted to the reality of the inclusive school paradigm, in general, and to the development of assertive and inclusive responses for students with specific needs, in particular.

Thus, it seems pertinent to us to reflect and review teacher training, namely in its structuring and organization of curricular units with regard to the dimensions of time, spaces, and reality or contexts where they should be developed.

In short, what was found in this study suggests the need for reflection on the part of political decision makers in the area of education and, very particularly, teacher training institutions. In this sense, it would be pertinent to consider the need to consider the reinforcement in study plans of specific long-term curricular units. These would aim not only to acquire theoretical and practical knowledge, but particularly to enable the consolidation of this same knowledge. At the same time, it would also enable the development of learning practices in a real context. In this context, there would not only be direct contact with students with disabilities, but, particularly, the development of guided and supervised pedagogical practices.

Likewise, it would be important to reflect on the abandonment of a vertical training logic, based on one or two specific curricular units, for a horizontal training dynamic, where teacher training for special educational needs is addressed in a transversal way in all the curricular units of the course.

These assumptions, in turn, encourage us to carry out future research, namely answering the following questions: "What is the profile of the inclusive physical education teacher?", "What impacts would it have on initial training in specific contexts of inclusion?", and "What would be the impact of the transversality of contents and specific competencies in special educational needs, by the different disciplines throughout the training processes, in improving the efficiency of teacher training?".

**Author Contributions:** T.C.: Conceptualization, methodology, software, validation, formal analysis, investigation, resources, writing—original draft preparation, funding acquisition; E.R.: Methodology, investigation, visualization, project administration, funding acquisition; E.G.M.: investigation, visualization, supervision, writing—review and editing; L.L.: investigation, visualization, supervision, writing—review and editing; A.P.: methodology, software, validation, formal analysis, investigation,

resources, data curation, writing—review and editing, visualization, project administration, funding acquisition. All authors have read and agreed to the published version of the manuscript.

**Funding:** This work is funded by National Funds through the FCT—Foundation for Science and Technology, I.P., by project reference UIDB/05507/2020 and DOI identifier 10.54499/UIDB/05507/2020. Furthermore, we would like to thank the Centre for Studies in Education and Innovation (CI&DEI) and the Polytechnic of Viseu for their support.

**Institutional Review Board Statement:** The study was conducted in accordance with the Declaration of Helsinki.

**Informed Consent Statement:** Informed consent was obtained from all subjects involved in the study.

**Data Availability Statement:** All data generated in the project can be obtained by contacting the corresponding author.

**Conflicts of Interest:** The authors declare no conflict of interest.

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
