# Peer review of "Physical Education Teachers’ Representations of Their Training to Promote the Inclusion of Students with Disabilities"

_education, doi:10.3390/educsci14010049_

Round 1

Reviewer 1 Report

Comments and Suggestions for Authors

Abstract:

The abstract provides a clear overview of the study's objectives and findings. It highlights the importance of inclusive education, particularly in the context of Physical Education (PE), and emphasizes the role of teacher training. However, there are areas where it could be improved:

·         Clarity and Conciseness: While the abstract is informative, it could benefit from more concise language to make it easier to grasp quickly. Consider breaking some of the lengthy sentences into shorter, more digestible ones.

·         Specificity: The abstract mentions "Specific Health Needs (SES)" without explaining this term. Providing a brief definition or clarification would be helpful, as not all readers may be familiar with the acronym.

·         Results and Implications: The abstract briefly mentions the study's findings but could provide a bit more detail about the key results. Additionally, it could include a sentence or two about the practical implications of these findings for teacher training and inclusive education in PE.

·         Engagement: To make the abstract more engaging, you could mention the potential benefits of improving teacher training for inclusive education and how it contributes to the broader goals of inclusive schooling.

Introduction:

The introduction provides a comprehensive overview of the study's context and objectives. It discusses the importance of inclusive education and the role of teacher training in achieving it. However, there are areas where the introduction can be improved:

·         Clarity and Conciseness: The introduction contains long sentences that might make it challenging for readers to follow the main points. Consider breaking down some of the sentences for clarity and ease of reading.

·         Specificity: While the introduction mentions "Special Health Needs (SES)," it would be helpful to provide a brief explanation or definition of this term to ensure all readers understand its significance.

·         Engagement: To make the introduction more engaging, you could incorporate a brief statement about the significance of understanding PE teachers' perceptions of training in the context of inclusive education and the potential impact on inclusive schooling.

·         Structure: The introduction presents numerous aspects related to teacher training, legislative changes, and teacher perceptions. It might benefit from a clearer structure to guide the reader through these various components.

Materials and Methods:

The Materials and Methods section provides an overview of the participant selection, instruments, procedures, and data processing. Here's an evaluation of this section with some suggestions for improvement:

·         Participant Selection: The authors describe the participant selection process clearly and provide essential details about the participants, including their qualifications, teaching positions, academic background, and experience. This information helps readers understand the sample's characteristics.

·         Instruments: The section describes the development and validation of the questionnaire, which is crucial for understanding the data collection process. However, there's no mention of the questionnaire's content or specific questions. It would be helpful to provide a brief summary of the key questions or themes covered in the questionnaire.

·         Structure of the Questionnaire: The section provides a detailed breakdown of the questionnaire's structure, including the different parts and the topics covered in each section. This transparency is essential for readers who may want to replicate the study.

·         Procedures: The procedures for data collection are well-documented. It's clear how participants were contacted and how they accessed the questionnaire. The mention of obtaining informed consent is essential for ethical considerations. However, it could be beneficial to include information about the response rate or any incentives offered to participants.

·         Data Processing: The section outlines the approach to data analysis, involving both quantitative and qualitative techniques. It mentions the use of descriptive statistics and content analysis, which aligns with the mixed-methods design. However, it lacks details about the specific statistical tests or software used for quantitative analysis.

·         Citation of References: When referring to previous literature (e.g., Campos et al., 2015; McGrath, Crawford, & O'Sullivan, 2019; Pocock & Miyahara, 2018), it's important to cite these references properly within the text, providing readers with the sources for further reading.

·         Clarity and Conciseness: While the section is detailed, some sentences could be more concise, maintaining clarity while reducing wordiness.

Results:

The Results section presents the findings of the exploratory study, focusing on PE teachers' representations of their training for the inclusion of students with Special Health Needs (SES). Here's an evaluation of this section with suggestions for improvement:

·         Clarity of Objectives: The objectives of the study are clearly outlined, indicating a focus on two dimensions: initial training for teaching inclusive physical education and continuous training for inclusion. This clarity helps readers understand the scope of the research.

·         Data Presentation: The section effectively presents the data, using tables to organize and present the results. The use of tables makes it easier to comprehend the findings. However, tables should be consistently labeled and numbered for easier reference.

·         Clarity of Data Interpretation: The section interprets the data and provides insights into the teachers' perceptions and experiences. The descriptions of the findings are clear and directly related to the research objectives.

·         Quantitative Data: The presentation of quantitative data in tables is well-structured. The tables include percentages and numbers of participants for each category, enhancing the transparency of the findings. It's also useful that tables include categories for "Does not reply" and "No opinion" to account for missing data.

·         Qualitative Data: The presentation of qualitative data in tables is informative. However, it could benefit from more detail about the specific subcategories mentioned in the tables. This will provide a clearer picture of teachers' suggestions and reasons for continuous training.

·         Proper Citation: The section appropriately references the tables, making it easy for readers to connect the text to the corresponding data. However, it could be improved by providing a brief contextual description for each table when first mentioned in the text.

·         Suggested Improvements: The section effectively conveys teachers' perceptions of their initial and continuous training for inclusion, highlighting the need for improvement. Furthermore, it identifies that teachers generally don't feel well-prepared to work with students with SES.

Discussion:

The Discussion section of the article addresses the findings regarding the training and preparedness of physical education (PE) teachers for inclusive education, with a particular focus on students with Special Educational Needs (SEN). Here is an evaluation of this section with suggestions for improvement:

·         Clarity of the Argument: The discussion opens by acknowledging the central role of teacher training in inclusive education. It establishes the importance of PE teachers' training for the success of students with SEN. This provides a clear context for the subsequent analysis.

·         Citation and Integration of Previous Studies: The section effectively integrates findings from previous studies to support the presented data. This contextualizes the current study within the broader academic conversation on the topic, strengthening the argument. However, it would be beneficial to include specific citations in the text to give proper credit to these previous studies.

·         Specific Data Interpretation: The discussion presents the key findings related to teachers' perceptions of their initial training. It interprets these findings effectively, highlighting that initial training falls short in preparing teachers for inclusive education. This interpretation is supported by references to other studies.

·         Analysis of Competencies: The discussion provides a valuable analysis of the competencies in which teachers felt more prepared after their initial training. This analysis helps to understand why teachers might feel unprepared for inclusive education, as the focus was on traditional academic knowledge rather than practical inclusive strategies.

·         Analysis of Contact with Students with SES: The discussion draws attention to the lack of contact with students with SES during teachers' initial training, which aligns with the challenges identified in the study. This emphasizes the gap in practical experience.

·         Use of Statistics and Quotes: The discussion benefits from the use of statistics, such as percentages, to support arguments. However, it could be further enhanced by incorporating direct quotes or responses from the teachers, which would provide a more tangible connection to the participants' voices.

·         Interpretation of Continuous Training: The section also discusses the significance of continuous training in the context of special educational needs, emphasizing its role in facilitating inclusive PE. It connects the demand for such training to teachers' need for practical interventions.

·         Critique of Existing Training Models: The discussion critically evaluates the current training models, which seem to be inconsistent with the needs of teachers. This critique highlights the issues with traditional training models that focus on theory rather than practical application.

·         Informed Autonomy: The discussion mentions that PE teachers autonomously seek their own knowledge. This concept is intriguing and could be expanded upon to explore how teachers' self-initiated learning might impact their effectiveness in inclusive PE.

Conclusions:

The Conclusions section provides a summary of the key findings and insights from the study. Here's an evaluation of this section with some suggestions for improvement:

·         Clarity of Conclusions: The section presents clear and concise conclusions that directly relate to the study's findings. It effectively highlights the imbalance between declarative knowledge and procedural knowledge in teacher training and the need for improvements.

·         Integration of Findings: The conclusions reflect the main findings presented in the earlier sections. This alignment ensures that the conclusions are grounded in the study's data.

·         Call for Reflection and Review: The section appropriately calls for reflection and review of teacher training, particularly in terms of structuring and organization. It emphasizes the need to rethink curricular units and the training approach to better prepare teachers for inclusive education.

·         Potential Future Research: The conclusion raises relevant questions for future research, including exploring the profile of inclusive PE teachers and the impact of a more transversal approach to special educational needs in teacher training. This highlights a research agenda for further investigation.

·         Conciseness: The section is concise and to the point. It avoids unnecessary repetition or restatement of the findings.

·         Missing Recommendations: While the section identifies areas for reflection and future research, it could benefit from explicit recommendations for policymakers, educational institutions, and teacher training programs on how to address the highlighted issues. Providing practical recommendations would make the conclusions more actionable.

·         Clear Structure: The section has a clear and logical structure, with each paragraph building on the previous one. The flow of ideas is well-organized.

Author Response

We would like to thank the reviewers for their suggestions for improving the manuscript, which we have indicated in red in the body of the text (for better identification) and which we have listed below by area / topic: 

Changes were made to the "Abstract" item (reviewer 1 and 2);
Changes were made to the "Introduction" item (reviewers 1, 2 and 3);
Changes were made to the item "Material and methods": 1. Participants; 2. Instruments; 3. Procedures; 4. Data processing (reviewer 1);
Changes were made to "Results": 1. Initial training for teaching in inclusive physical education; 2. Continuous training for inclusion (reviewers 1, 2 and 3). 
Changes were made to the "Discussion" item (reviewers 1, 2 and 3);
Changes were made to "Conclusions" (reviewers 1 and 2).
A general revision of the English was carried out (reviewer 3).  
In addition, we found that the DOI references were incorrect. All of these situations have been rectified.
The authors.

Reviewer 2 Report

Comments and Suggestions for Authors

Dear Authors,

It is a pleasure to review your article titled "Physical education teachers' representations of their training to promote the inclusion of students with disabilities," focusing on the training of Physical Education (PE) teachers in Portugal for including students with Specific Health Needs. I would like to share some observations that I believe could significantly enrich your work.

One of the main aspects to consider is the analysis of the results. The data presentation, particularly in sections discussing survey responses from teachers, lacks the depth and analytical rigor expected in a study of this nature. The descriptive approach limits the ability to offer significant insights and leads to conclusions that appear overly generalized and not sufficiently supported by a detailed analysis.

Additionally, the methodology used, particularly the use of Google Forms for data collection, presents multiple limiting factors. These include potential lack of sample representativeness, limitations in response validation, challenges in ensuring the authenticity of participants, and difficulties in managing response variability. These methodological limitations should be discussed more thoroughly to provide a clearer understanding of how these factors might have impacted the study's results and conclusions.

In terms of writing, the article would benefit from greater conciseness and clarity, especially in the abstract and introduction, where inconsistent terminology is observed. A more detailed explanation of participant selection and questionnaire design in the methodology section would significantly improve the reader's understanding of the decisions made during the research.

It would also be valuable to more clearly contextualize the findings within the international context of inclusive education and to more robustly justify the research gap that your study addresses. This could include comparisons with similar studies in other countries.

The practical implications of your findings could also be made more explicit. Concrete recommendations for teacher training in similar contexts would increase the practical value of your research. Moreover, a deeper exploration of the limitations and suggesting specific directions for future research would strengthen your contribution to the field.

In summary, although your article offers valuable perspectives, it would greatly benefit from increased depth and rigor in its analysis, clarity in writing, and a more detailed discussion on methodological limitations and their implications.

I appreciate the opportunity to review your work and hope these suggestions will be useful. I am confident that integrating these changes will significantly enrich your study and its impact in the field of inclusive education.

Sincerely,

Author Response

(The authors gave the same response as above.)

Reviewer 3 Report

Comments and Suggestions for Authors

Thank you for giving me this opportunity to review the manuscript “Physical education teachers' representations of their training to promote the inclusion of students with disabilities” Before this study can be published, below are my suggestions for revisions. 

Introduction:

Overall, please rewrite the introduction. I found it challenging to find the primary objective the authors aimed to convey. Some information presented in the introduction section is confusing. The authors should provide more contexts for readers. For example: 

1.     What is “the Sustainable Development Agenda 4 (SDG 4)?” Please explain the context. 

2.     “Some guidelines have emerged with an impact on teacher training programs and, very particularly, on their training to act in the context of the inclusive education paradigm [3, 4], with the consequent need to train professionals capable of assuming responsibility for the education of all students without exception [3, 4], through a greater prior awareness of the importance of the inclusive philosophy for the development of societies.” I would suggest the authors rewrite this. The authors should break this lengthy sentence into separate sentences for improved clarity.

3.     “From the need to define an educational policy articulated between the various countries of the European Union, the basic guidelines for the harmonization of higher education systems and, consequently, the reform of teacher training emerge from the Bologna Declaration.” I do not understand this. Please edit it to improve clarity and provide more contexts for readers. Is related to the current study? If it is relevant, please provide a more explicit analysis in reference to the focus of the present study.

4.     What is the purpose of including the information based on legislation? Is related to the current study? If it is relevant, please provide a more explicit analysis in reference to the focus of the present study.

5.     Please consider reviewing more existing literature. The authors need to do more in the literature review to analyze the current challenges and gaps. 

6.     Where is the research question section? What are the research questions? 

Methods: 

1.     “Of these teachers, 42% 94 (n=64) taught in the 2nd cycle and 58% (n=87) in the 3rd cycle and secondary.” What does cycle mean? 

2.     I saw SES stands for Specific Health Needs in the summary. However, the authors should define and describe this population in the introduction. The information is completely missing in the intro, and suddenly the term “SES” came up in the method section. Very confusing. 

3.     The questionnaire is key in the present study. Please consider using bullet point to list each part of the questionnaire and add more information to describe each part of the content. What does it mean initial training? What does it mean continuous training? And why did the authors design the questions? 

Results: 

1.     Since there is no research question, I have a hard time understanding the results section. 

2.     In the method section, the authors described how they analyzed qualitative data, but I did not find qualitative results in the results section. 

Discussion:

1.     The authors have compared their results with findings from previous research studies. To enhance comprehension and make the comparisons more meaningful, the authors should expand on the findings of these previous studies in the introduction. This will provide readers with the necessary context.

2.     I recommend that the authors incorporate practical implications for practitioners to guide inclusive practices in physical education classes.

3.     Additionally, I noticed the frequent use of "SES," "SEN," and "NSE." It's unclear if these are typos or abbreviations, but consistency and clarification should be maintained. It would be beneficial to review the text for any writing errors.

4.     Given the challenges in understanding the current writing, I suggest considering English editing services to improve the overall clarity and readability of the text.

Comments on the Quality of English Language

 English very difficult to understand/incomprehensible

Author Response

(The authors gave the same response as above.)

Round 2

Reviewer 2 Report

Comments and Suggestions for Authors

Dear authors,

Thank you for your efforts to improve your work. After a detailed reading of the new paper, I am sorry that the discussion does not address the explanation of your results in a correct and profound way. 

It would be interesting to deepen the discussion to a greater extent taking into account the objectives and hypotheses of the study.

With those improvements, in my opinion, the article could be publishable.

Thank you for your efforts,

Author Response

The authors would like to thank the reviewer for his comments. We have therefore rewritten this section to make it more robust.

P.S.: all new changes made are highlighted in blue. 

Reviewer 3 Report

Comments and Suggestions for Authors

The authors have addressed all of my comments. The manuscript should be ready for publication. 

Author Response

We are extremely grateful for your contributions and your trust in our work.